# Exploring the Interplay of the Physical Environment and Organizational Climate in Innovation

**Lei Peng** [1,2] **and Ruiying Jia** [1,2,*]

1    School of Architecture & Urban Planning, Huazhong University of Science and Technology, Wuhan 430074, China; penglei@hust.edu.cn

2    Hubei Engineering and Technology Research Center of Urbanization, Wuhan 430074, China

*    Correspondence: ruiyingjia@hust.edu.cn

**Abstract:** The intricate relationship between physical and social environments within organizations plays a pivotal role in shaping innovation endeavors. This paper introduces a three-dimensional framework aimed at comprehending the intricate mechanisms through which the physical and social environments synergistically drive innovation. Building on this, a systematic four-dimensional framework (communality, individuality, comfort, and health) is proposed to structure a comprehensive literature review, mapping out the intricate linkages between innovation and the physical environment. Through this extensive review, we delve into the intricate connections between the physical innovation environment and the broader innovation climate, unearthing valuable insights. Additionally, we highlight two promising directions for future research within the realm of physical environment–innovation climate interactions. Furthermore, we underscore the paramount importance of embracing an interdisciplinary approach, seamlessly blending perspectives from both the physical and social spheres to gain a holistic and nuanced understanding of the innovation landscape. This integrated viewpoint is pivotal to unraveling the multifaceted dynamics that underlie successful innovation initiatives.

**Keywords:** innovation; physical environment; innovation climate

## 1. Introduction

The evolution of contemporary economic development, shifts in work styles, and the imperative for innovative production have triggered organizational changes, impacting both workspace design and organizational structures. Enterprises now face the continual need to innovate their products or services to bolster core competitiveness. Consequently, the strategies for fostering corporate innovation and sustainable development have emerged as a pivotal research area in the twenty-first century. Initial research concentrating on individual employee creativity within organizations bears certain limitations (Styhre & Sundgren, 2005 [1]). However, a prevailing focus in contemporary organizational research revolves around investigating environmental elements that exert influence on employee innovation at the organizational level. Given the intricate nature of organizational surroundings, divergent viewpoints on the relationship between innovation and the environment emerge across various disciplines. Despite the extensive historical research delving into innovation through the lenses of organizational sociology and architecture, the amalgamation of these perspectives to explore innovation remained a relatively unexplored avenue until recent times (Lukersmith & Burgess-Limerick, 2013; Blomberg & Kallio, 2022) [2,3].

Exploring the promotion of sustainable innovative organizational environments necessitates the delineation of three distinct trajectories within academic research: innovation behavior, the physical spatial organization within organizations, and the ambient milieu within these establishments. The disciplinary demarcations that separate sociology

and architecture impose challenges on scholars within each domain, hindering a holistic comprehension of organizational environments. Despite architecture's comprehensive scrutiny of office environments (Vilnai et al., 2005) [4], it falls short of investigating the attributes of physical spaces conducive to innovation, lacking a systematic framework akin to that in sociology. Our objective is to furnish theoretical augmentation for designing office spaces that foster innovation behavior. We also advocate for parity in the attention devoted to both physical space and operational management in the realms of research and the practical implementation of innovation, a call directed towards scholars and entrepreneurs alike. Consequently, this paper's analysis predominantly gravitates toward the realm of architectural research. We intend to comprehensively review and encapsulate the cross-pollination of the physical spatial environment and the atmosphere conducive to sustainable innovation. The research content encompasses the following areas:

1.  We establish a three-dimensional analytical framework for the social environment, the physical environment, and innovation to sort out the intersection between the three of them (Figure 1).
2.  Based on the four elements of the physical environment proposed in this paper, the existing literature on innovation and the physical environment is reviewed.
3.  Based on the literature review, the relationship between the physical environment of innovation and the climate of innovation is explored.
4.  Two future research directions are proposed to guide future research on integrated innovation environments for the physical and social environments.

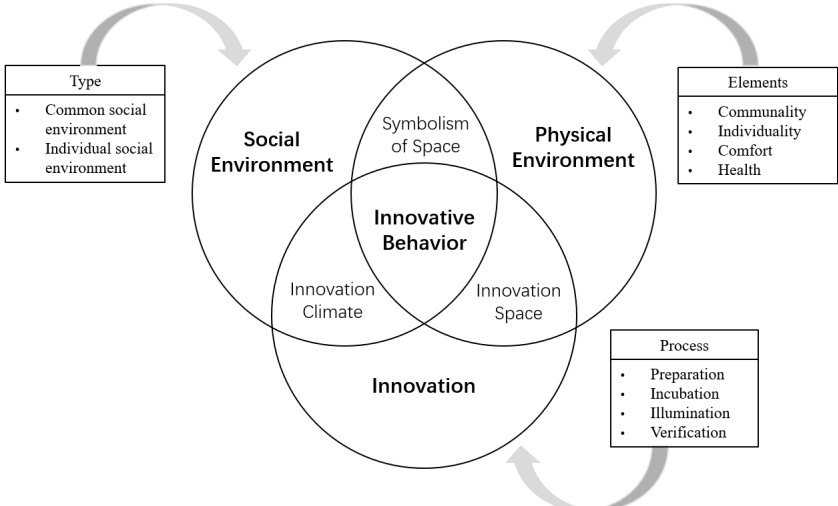

**Figure 1.** A three-dimensional framework for analyzing innovative behavior.

## 2. Researching the Challenges of Innovation Environments

This article centers its attention on innovation within the organizational setting, parsing the organizational environment into two distinct components: the physical environment and the social environment (Morton et al., 2016; Billie and Robert, 2002) [5,6]. Based on social cognitive theory (Bandura, 1986) [7] and environmental behavioral theory (Moudon A V and Lee C, 2003) [8], we meticulously dissect the physical and social environments from the vantage points of individual behavior and perception. The subsequent review and discussion offer valuable insights and practical guidance in navigating this multifaceted landscape.

### 2.1. Innovation and Creativity

The term "innovation" has been extensively explored from a multidisciplinary perspective. Scott and Bruce (1994) [9] assert that innovation constitutes a process encompassing both idea generation and implementation. In research, "innovation" and "creativity" are

often utilized interchangeably. Amabile et al. (1996) [10] propose that individual and collaborative creativity serve as the foundational bedrock for innovation. Winks et al. (2020) [11] contend that creativity acts as both the precursor and outcome of the innovation process. While creativity pertains to the formulation of original and useful concepts (Mumford and Gustafson, 1988) [12], innovation revolves around the adoption and realization of these novel and valuable ideas (Kanter, 1988) [13]. It is imperative to recognize that innovation and creativity are symbiotic, mutually reinforcing elements, differing in emphasis more than they do in substance (West & Farr, 1990) [14]. Therefore, the definition of innovation adopted within this paper encompasses the entirety of the process involving the generation and implementation of novel ideas, or the creation of novel entities within an organizational context, inherently encapsulating creativity as an integral component.

Creative thinking, the bedrock of innovation and creativity, forms the cornerstone of sociological inquiry into innovation. Wallas's (1926) [15] seminal Four-Stage Model of the Creative Process, comprising preparation, incubation, illumination, and verification, significantly advanced the study of creative thinking in psychology and systematically delved into the innovation process. Building upon this framework, this paper aligns with Scott and Bruce's (1994) [9] definition of innovative behavior, which characterizes it as the actions individuals undertake in the pursuit of innovation. This is further delineated into three stages: (1) identifying a problem and generating ideas or solutions; (2) seeking support for one's ideas; and (3) developing innovative standards or models that can be disseminated, produced on a large scale, and subsequently utilized to actualize one's innovative concepts. Kleysen and Street (2001) [16] consolidate the existing literature, identifying five stages of personal innovation: opportunity seeking, idea generation, investigation, support, and application. They conceptualize personal innovation as the genesis, introduction, and application of valuable ideas across all levels of organizational behavior. This comprehensive perspective provides a robust framework for understanding and advancing innovation within the societal context.

### 2.2. Commonly Perceived Social Environment

Repetti (1987) [17] categorized the work social environment into common and individual social environments. This paper focuses on the analysis of the common social environment, which means that the social climate is shared by employees in the same work setting.

### 2.2.1. Organizational Climate Environment

Starting in the 1960s, organizational climate has gained increasing attention from organizational researchers as a critical aspect of the human social environment. Through the lens of the shared cognitive approach, there is a consensus among researchers that organizational climate pertains to members' perceptions or experiences of the prevailing organizational environment (James et al., 2008) [18]. It represents a systematic organizational attribute formed by the collective perceptions of members about the organizational environment, ultimately influencing their behaviors (Sleutel, 2000) [19]. Numerous studies have demonstrated the significant impact of organizational climate on employee behavior and psychology (Anderson & West, 1998; James et al., 2008) [18,20]. The diverse nature of organizational climate dictates that different climate elements wield varying effects on employee behaviors. Schneider et al. (2013) [21] posit that organizational climates can be categorized into result-oriented and process-oriented climates based on their core orientations. Researchers investigating climate often center their attention on specific policies, practices, and procedures as the sources of individuals' perceptions. They delve into how employees perceive the outcomes of the organization's management (e.g., service quality, safety, and innovation) and the corresponding internal processes (e.g., fairness, ethics, and inclusivity). It is contended that climate serves as behavioral evidence reflecting the cultural attributes of the work environment. These behaviors, in turn, constitute the foundation upon which employees formulate their perceptions of the organization's values and beliefs.

2.2.2. Innovation Climate

The concept of innovation climate entails exploring the connection with innovation building upon the study of organizational climate, which significantly contributes to enhancing employee creativity, organizational efficiency, and competitiveness. According to Schneider et al.'s (2013) [21] classification of organizational climate, innovation climate research is positioned as an investigation centered on organizational innovation as a strategic outcome. In this study, we delve into highly cited quantitative analyses of innovation climate (Table 1), operating at the level of shared perceptions, and categorize them into team and organizational levels based on the research subjects' focus.

At the organizational level, for instance, Amabile et al. (1996) [10] define organizational innovation climate as the collective perceptions held by organizational members regarding the presence of an innovative environment within the organization. To assess this climate, Amabile et al. employ the KEYS (Innovation Climate Evaluation Scale) to quantitatively examine elements of the organizational environment conducive to fostering innovation, such as the promotion of challenging goals and recognition for creative work. Scott and Bruce (1994) [9] propose a model where individual innovative behavior arises from the interplay of four factors: the individual, the leader, the work group, and the organizational innovation climate. Empirical research conducted on employees of R&D companies by Scott and Bruce revealed that the dimension of innovation support within the organizational climate significantly influences individual innovative behavior. On the team level, Anderson and West (1998) [20] introduce the Team Climate Inventory (TCI) scale, comprising four dimensions: safety, support for innovation, willingness, and task orientation. They further break down safety into the security of participation and frequency of interaction with five dimensions. A majority of innovation climate research scales are adapted versions of these quantitative studies (Newman et al., 2019) [22]. In this paper, we adopt Schneider et al.'s (2013) [21] classification of organizational climate and categorize elements from the aforementioned quantitative research on innovation climate into two orientations: those focused on stimulating the innovation process (e.g., encouragement and safety) and those aimed at realizing innovation (e.g., rewards, challenges, and tasks). This categorization prepares the groundwork for exploring the associations between atmosphere and space in subsequent sections.

**Table 1.** Highly cited quantitative analyses of innovation climate.

| Author | Innovation Climate Measure | Dimensions of Innovation Climate | Climate Orientation |
|---|---|---|---|
| Scott & Bruce (1994) [9] | Psychological Climate for Innovation | Support for innovation; Resource supply | process-oriented |
| Amabile et al. (1996) [10] | Assessing the climate for creativity | Encouragement of creativity; Autonomy or freedom; Resources; Pressures; Organizational impediments to creativity | process-oriented |
| Tesluk (1997) [23] | Innovation Climate Assessment Scale | Goal emphasis; Means emphasis; Reward orientation; Task support; Socioemotional support | process-oriented & result-oriented |
| Anderson & West (1998) [20] | Team Climate Inventory | Vision; Participative safety; Task orientation; Support for innovation; Interaction frequency | process-oriented & result-oriented |

### 2.3. Physical, Social, and Organizational Environments

Human behavior is not solely propelled by intrinsic factors like motives and attitudes; it can also be influenced by impromptu actions triggered by the surroundings. Social cognitive theory underscores the dynamic interplay between the individual and their environment. Within this framework, the components of the individual, their behavior, and the environment coexist independently while simultaneously interconnecting and shaping one another (Bandura, 1986) [11]. In this paper, the organizational environment is divided into the objective physical environment and the social environment (Morton et al., 2016; Billie & Robert, 2002) [5,6]. While achieving a comprehensive understanding of organizational environments demands interdisciplinary investigation, research into physical spaces frequently remains siloed and explored within specific disciplines that extend beyond the customary realms of organizational behavior and management. These encompass fields like architecture, environmental psychology, facilities management, and education (Brown et al., 2005; Orlikowski, 2010) [24,25]. The following section explores the influence of the physical and ambient environments on behavior and perception, respectively, and discusses the similarities between the two that influence behavior, providing a basis for summarizing the discussion of the innovation climate and spatial associations.

### 2.4. Four-Dimensional Framework of the Physical Environment

The physical environment is defined using Stephenson et al. (2020) [26] as the built environments that emerge from organizational activities, objects, arrangements, and social practices. Makela et al. (2018) [27] introduced a comprehensive framework for designing the physical environment. Among these, communality aligns with individuality, while comfort corresponds with health. Additionally, they emphasized the significance of the first four elements through a post-use assessment of practical design ventures. Achieving a balance between communality and individuality is crucial for optimizing space utilization and human behavior. Likewise, the physical environment must cater to both human comfort and health perceptions. Moreover, striking a balance between evolving needs and the physical environment is pivotal for incorporating novelty while respecting tradition. This paper maintains a steadfast focus on human behavior and perception within physical spaces, a vital foundation for seamlessly integrating the physical and social environments in subsequent exploration. As a result, the physical attributes of the work environment are categorized into four fundamental elements: communality, individuality, comfort, and health (Table 2). This classification serves as a robust framework for further investigation and implementation.

**Table 2.** Four-dimensional framework of the physical environment.

| Physical Environment Elements and Behavioral Perception | | | | |
|---|---|---|---|---|
| **Element** | **Communality** | **Individuality** | **Comfort** | **Health** |
| Sub-element | Open and transparent, close layout distance | Enclosed, adjustable partitions | Spaciousness, comfortable furniture, esthetic colors, light, natural landscape, acoustics | Air quality, open space |
| Behavior/Perception | Communication, cooperation | Individual work, focus, free setup | Visual comfort, comfort of use | Air health, rest, and recovery |

In terms of behavior, the communality and individuality nature of the physical environment refers to the effect of varying degrees of openness and accessibility of space on behavior. Researchers have focused on the communal attributes of space in terms of communication and interaction between employees (Blomberg and Kallio, 2022) [3]. Hall



(1966) [28] suggests that the physical environment can influence the behavior of human interactions through scales that affect individuals differently. Allen (1977) [29] was among the first to quantitatively establish a connection between spatial components and social behavior. His research highlighted that as the distance between workstations increased, the frequency of communication decreased. This pioneering study by Allen catalyzed increased scholarly interest in elucidating the mechanisms that interconnect the public attributes of the physical environment with communication and collaboration (Eric & Mary 1986 [30]; Hatch, 1987 [31]; Lile et al., 2009 [32]; Wineman et al., 2009 [33]; Salazar and Claudel, 2022 [34]). The personalization of space pays more attention to the impact of personal-scale spatial elements such as workstations on the individual's independent work (Ainsworth et al., 1993; Jiang et al., 2021; Ko et al., 2020) [35–37].

From the perception perspective, comfort refers to an individual's perceived level of comfort with the physical environment, such as space, furniture, and light, while health refers to indoor air quality and space for rest and recovery. The comfort of a space can be evaluated from two distinct angles. Firstly, cognitive comfort encompasses the sense of spaciousness within the environment (Zhuang et al., 2022) [38] as well as the esthetic qualities of its decor (Ainsworth et al., 1993) [35], both of which contribute to heightened job satisfaction. Secondly, physical comfort pertains to ergonomically designed furniture, which has been substantiated by research to enhance employees' efficiency. The healthiness of space primarily manifests in the caliber of the indoor environment, encompassing factors like lighting, temperature, humidity, and ventilation (Ko et al., 2020; Khoshbakht et al., 2021; Zhuang et al., 2022) [37–39]. These aspects exert an influence on employees' physical sensations, which can indirectly affect employee productivity or organizational performance. With the emphasis on exercise, researchers have begun to look at how workplace layout design can promote physical activity to improve employees' physical health (Candido et al., 2019) [40].

### 2.5. Symbolism Connecting the Social and Physical Environments

Rafaeli and Vilnai-Yavetz (2004) [41] proposed that one of the characteristics of the physical environment is symbolism, particularly its organizational culture. This means that the way a workspace is designed, arranged, and decorated can communicate important aspects of the organization's values, norms, and identity. In sociological research, organizational climate and organizational culture are inextricably linked, with climate being the basic vehicle of culture, encompassing the intangible aspects of culture, which are manifested through work processes, organizational goals, and a range of observable behaviors that contribute to a "tangible" climate (Ahmed, 1998 [42]; Chan, 1998 [43]; Schneider et al., 2002 [44]). Thus, organizational climate, as a common social environment, can "tangibly" express the organizational culture symbolized by physical space, effectively linking the two (Figure 2).

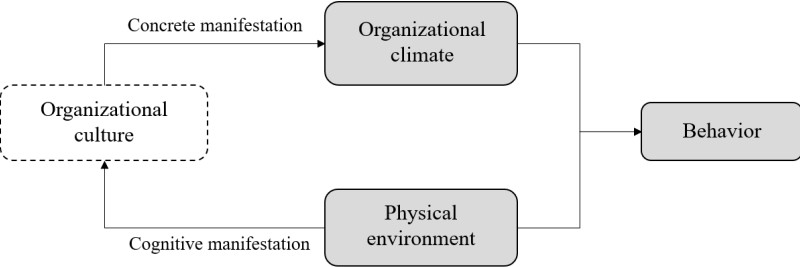

**Figure 2.** Relationship between the three elements that influence behavior.

To grasp innovative behavior within an environment, it is essential to initially comprehend the interplay between the environment and behavior. Consideration of the multidisciplinary effects of the social and physical environments on behavior is currently underappreciated. Frank (1984) [45] argued that the environment's influence on behavior

is not singular and direct but rather a convergence of physical and social factors that collectively shape it. Olikowsik (2010) [25] contended that the conventional realm of management overlooks the intricate connection between organizations and the tangible spaces that underlie human actions and interactions. He advocated for the simultaneous exploration of physical space and organizational environments, shedding light on the intricate ways in which society and matter interact in everyday life.

Present perspectives on the interplay between behavior and social and physical environments are categorized into two primary streams. The first perspective underscores the intricate entwinement of physical and social environments. Billie and Robert (2002) [6] argued that the physical environment serves as a necessary support for behavior, ranking second only to personal and social determinants in influencing behavior. Lukersmith and Burgess (2013) [2] investigated healthcare workers' creativity by considering job content and leadership as social variables and interior decoration, sound, light, and heat as physical variables. They concluded that the social and physical environments collaboratively stimulate creativity, with the social environment wielding a more potent influence on creativity than the physical environment. However, some scholars contend that the physical and social environments exert separate and independent influences on behavior. For instance, Dul (2011) [46] examined the impact of the physical work environment on the creativity of knowledge workers. Through a questionnaire survey involving 274 knowledge workers, it was discerned that creative personality, social and organizational environment, and physical work environment distinctly and progressively influence creative performance independently.

The correlation between behavior and the two environments bolsters our proposed three-dimensional framework for the innovation environment, which is centered on innovation behavior. The objective is to unveil a comprehensive understanding of the connection between the physical and social environments within innovation contexts. Current literature has primarily concentrated on research concerning the link between the social environment, specifically the innovation climate, and innovation, overlooking the role of the physical environment. Consequently, we proceed to examine this overlooked component, followed by an exploration of the interplay between the social and physical environments, drawing from the three-dimensional innovation framework.

## 3. Method

The above summarizes the physical space and organizational climate that influence behavior and perception, respectively, and sorts out the current quantitative research perspectives on innovation-based social environments: innovation climate. Physical environments have been recognized by scholars in both theoretical and empirical research on innovation, but there is no clear description of the elements of the physical environment that affect innovation. In the subsequent sections, the authors aim to address these gaps. This review builds on the above summary of the four elements of the physical environment by reframing existing quantitative and qualitative findings on innovation and the physical environment.

The process of collecting literature on topics related to innovation and the physical environment (Figure 3) was divided into two stages. In the initial stage, a search was conducted within the Web of Science database for English literature up to 2023. Since innovation, as defined in this paper, includes the concept of creativity, another set of keywords included "innovation" and "creativity". Physical environments and spatial design are inextricably linked, so the keywords "physical environment", "physical space", and "spatial design" were chosen. Six keyword combinations, such as "innovation AND physical environment", "innovation AND physical space", and "innovation AND space design", were employed to retrieve relevant sources. Furthermore, numerous studies have substantiated the pivotal role of communication and cooperation in fostering innovation. Social behaviors like collaboration and serendipitous interactions are perceived as tangible indicators or catalysts of innovation (Allen, 1977 [29]; Toker and Gray, 2008 [47]; Wagner et al., 2011 [48]; Yubo et al., 2021 [49]). Consequently, this paper extends the search by

incorporating the additional keyword "collaboration communication". Then, we used the Web of Science database to search the English literature up to 2023. Keyword combinations, such as "physical space AND collaboration communication", "physical environment AND collaboration communication", and "space design AND collaboration communication", were utilized to comprehensively explore the literature landscape.

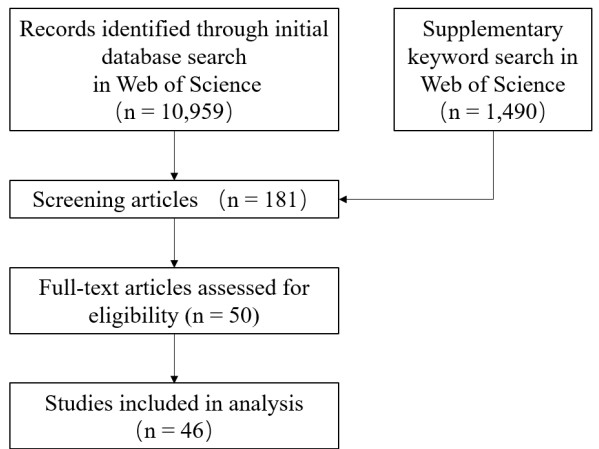

**Figure 3.** Flow diagram of searched, screened, and included studies.

By excluding research areas such as urban planning scales, a total of 181 research papers were identified. Through two rounds of manual screening, the titles, keywords, and abstracts of these papers were scrutinized to ascertain their relevance to either physical space and innovation or physical space and communication. The objective was to ensure that the literature under consideration explored innovation from a materiality standpoint. Ultimately, 46 papers were deemed suitable for analysis.

## 4. Result

### 4.1. Physical Environment and Innovation

The significance of the physical environment as a catalyst for innovation has garnered recognition within both theoretical and empirical inquiries within the innovation realm. First, our review consisted of combing through the currently available literature that summarizes the attributes of the physical environment for innovation (Table 3). Second, we explored the remaining literature by categorizing it according to the four elements of the physical environment summarized above and concluded with 12 secondary elements (Table 4).

**Table 3.** Studies of the elements of the physical environment that influence innovation.

| Author | Spatial Elements Affecting Innovation/Creativity |
|---|---|
| Rafaeli & Vilnai-Yavetz (2004) [41] | Instrumentality; Esthetics; Symbolism |
| McCoy (2005) [50] | Spatial organization; Architectonic details; Views; Resources; Ambient conditions |
| Moultrie et al. (2007) [51] | Geographic location; Scale; Real/Virtual; Flexibility; Design values and imagery; IT resources; Data and information; Modeling and visualization resources; Constraints; Evolution |
| Oksanen & Stahle (2013) [52] | Collaboration enabling; Modifiability; Smartness, Attractiveness; Value reflecting |
| Blomberg & Kallio (2022) [3] | Elements of workspace; Social dynamics of space |

**Table 4.** A framework of elements of the physical environment that influence innovation.

| Primary Element | Secondary Element |
|---|---|
| Communality | Layout proximity |
| | Openness |
| Individuality | Esthetics |
| | Visibility |
| | Controllability |
| | Indoor acoustics |
| Comfort | Spacious context |
| | Comfortable furniture |
| | Nature closeness |
| | Light |
| Health | Air quality |
| | Supply of recreational space |

### 4.1.1. Communality

Building upon the earlier definition and characterization of communal aspects in the physical environment, elements conducive to innovation encompass layout proximity and openness (Table 5). Layout proximity pertains to the spatial distance between areas, while openness refers to the extent to which workspaces are accessible to the public. Social behaviors like collaboration and serendipitous interactions are perceived as tangible indicators or catalysts of innovation (Allen, 1977 [29]; Yubo et al., 2021 [49]). Given that communication and collaboration are vital components of the illumination and validation phases of the innovation process, the public accessibility of the physical environment must facilitate efficient means of communication during this crucial stage.

**Table 5.** Communality of the physical environment affecting innovation.

| Primary Element | Secondary Element | Sources | Nature of Research | Physical Variables | Innovation Variables |
|---|---|---|---|---|---|
| Communality | layout proximity | Hatch (1987) [31] | quantitative | distance | interaction |
| | | Moultrie et al. (2007) [51] | qualitative | location | communication efficiency |
| | | Toker & Gray (2008) [47] | quantitative | proximity | consultation, innovation process |
| | | Winema et al. (2009) [33] | quantitative | distance | collaborative innovation |
| | | Ying et al. (2010) [53] | quantitative | distance | support for collaboration |
| | | Ying et al. (2011) [54] | quantitative | distance | collaboration perception |
| | | Kabo et al. (2014) [55] | quantitative | proximity | scientific collaboration |
| | | Bouncken & Aslam (2019) [56] | quantitative | distance | interprofessional communication |
| | | Yubo et al. (2021) [49] | quantitative | distance | unexpected encounters, interdisciplinary innovation |
| | | Salazar & Claudel (2022) [34] | quantitative | proximity | collaborative innovation |
| | | Sevtsuk et al. (2022) [57] | quantitative | distance | e-mail exchange |
| | | Yacoub & Haefliger (2022) [58] | qualitative | distance | collaborative innovation |
| | | Xia et al. (2022) [59] | quantitative | proximity | interdisciplinary communication |
| | openness | Hatch (1987) [31] | quantitative | obstacle | interaction |
| | | Toker & Gray (2008) [47] | qualitative | obstacle | consultation, innovation process |

The spatial delineation of layout proximity encompasses three dimensions: within the office, within the building, and between the office building and adjacent regional spaces. Within the office, Hatch (1987) [31], conducting observations and interviews within two technology companies, discovered that office employees situated farther away from the office entrance had fewer interactions. There is evidence to suggest that interprofessional interactions are more frequent among practitioners sharing the same office (Bouncken & Aslam, 2019) [56]. Within a building, research indicates that the distance between offices directly impacts collaborative innovation. Longer distances between offices are associated with a reduced likelihood of collaborative innovation (Winema et al., 2009) [33]. Moreover, a higher overlap of the shortest paths between researchers in various functional domains within a building corresponds to an elevated probability of collaborative innovation (Kabo et al., 2014) [55]. Salazar and Claudel (2022) [34] demonstrated that researchers operating within the same building exhibit an increased likelihood of engaging in collaborative innovation. Toker and Gray (2008) [47], in a case study, identified that a closer distance between offices and laboratories enhances the probability of face-to-face consultations, subsequently influencing innovation research. Sevtsuk et al. (2022) [57] showcased that reduced distances between offices correlate with a higher likelihood of email exchanges within the same building. Ying et al. (2010 [53], 2011 [54]) revealed that shorter distances between meeting rooms and workstations amplify the probability of collaboration. Yacoub and Haefliger (2022) [58] gleaned from interviews with employees across various firms on the same floor that closer common workspaces facilitate communication among employees not directly affiliated with the same firm, fostering the potential for collaborative innovation. Moreover, the urban surroundings around a building can also sway employees' innovative behaviors at work. Moultrie et al. (2007) [51] posited that the geographical location of the work environment influences employees' communication efficiency and innovation potential, both within the office and on the company's shop floor. Yubo et al. (2021) [49] highlighted interdisciplinary chance encounters as a mechanism to nurture innovation. Employing a spatial syntax approach to quantify the complexity of buildings within a university, their analysis disclosed that proximity in path links between teaching spaces, canteens, and accommodations heightens the likelihood of serendipitous encounters and exchanges. Xia et al. (2022) [59] introduced the concept of multidisciplinary innovation (MDI) and evaluated it through campus spatial organization networks and social networks. Their study found that the proximity of campus spaces positively influences interdisciplinary communication links.

Although the openness of space is an important factor in promoting communication and innovation, there is less research literature in this category. Hatch (1987) [31] found that the fewer the obstacles and the fewer the number and height of partitions, the higher the frequency of interactive activities. Toker and Gray (2008) [47] compared six labs with different spatial openness and revealed that face-to-face counseling was more frequent in labs with fewer obstacles and more openness.

### 4.1.2. Individuality

The aspects of individuality within the physical environment encompass esthetics, visibility, controllability, and indoor acoustics (Table 6). These factors, to a certain degree, impact an individual's mood, sense of privacy, and ability to concentrate. They necessitate a personalized approach tailored to the specific needs of different individuals. Concentration is paramount in the initial two phases of the innovation process. A thoughtfully personalized space serves to safeguard and amplify focus, thereby facilitating the preparation and incubation stages of the innovation process.

**Table 6.** Individuality of the physical environment affecting innovation.

| Primary Element | Secondary Element | Sources | Nature of Research | Physical Variables | Innovation Variables |
|---|---|---|---|---|---|
| Individuality | Esthetics | McCoy & Evans (2002) [60] | qualitative | indoor esthetics | creativity |
| | | Kelly (2002) [61] | qualitative | decoration | innovation |
| | | Haner (2005) [62] | qualitative | indoor esthetics | innovation process |
| | | Ceylan et al. (2008) [63] | quantitative | design esthetics | creativity potential |
| | | Dul & Ceylan (2011) [64] | quantitative | decorative colors | work creativity |
| | | Lukersmith & Burgess (2013) [2] | quantitative | decorative colors | creative potential |
| | | Crawford (2018) [65] | quantitative | decoration | innovative production |
| | Visibility | Peponis et al. (2007) [66] | quantitative | shared vision of neighborhood workspace | communication |
| | | Stryker et al. (2012) [67] | quantitative | visibility | communication and collaboration |
| | | Lukersmith & Burgess (2013) [2] | quantitative | visual obstruction | creative thinking |
| | | Bernstein & Turban (2018) [68] | quantitative | open workspace | face-to-face communication |
| | Controllability | Kristensen (2004) [69] | qualitative | workstation | first stage of the innovation process |
| | | Lukersmith & Burgess (2013) [2] | quantitative | free setup of workstation | creative thinking |
| | | Motalebi & Parvaneh (2021) [70] | quantitative | interior decoration | creative thinking |
| | Indoor acoustics | Clements-Croome (2006) [71] | qualitative | noise | creative thinking |
| | | Lukersmith & Burgess (2013) [2] | quantitative | acoustics | creative thinking |
| | | Martens (2011) [72] | qualitative | noise | creativity |

The decor of the physical environment is a reflection of the unique attributes of the organizational context. The appearance of the environment is important because it reflects the values and norms of people and organizations (Kelly, 2002) [61] and is the substance that is most intuitively perceived by people through vision. An attractive work environment can inspire and stimulate innovation among employees in an office environment (Haner, 2005) [62]. Creating a creative appearance is the main motivation for designing a creative office, which can increase the motivation of office workers, which in turn improves productivity (Crawford, 2018) [65]. For optimal innovation stimulation, space design should be customized to cater to different activities, cognitive intensities, and personal inclinations (Martens, 2011) [72]. McCoy and Evans (2002) [60] conducted research into the impact of visually perceivable interior design elements—such as materials, colors, and shapes—on creativity, utilizing a comparative quasi-experimental approach. Their study unveiled that the utilization of more natural materials, incorporation of natural environmental components, and minimal usage of cool colors and artificial materials can foster employee creativity. Similarly, Ceylan et al. (2008) [63] demonstrated that well-executed office interior design has the potential to stimulate employee creativity. Colors, albeit indirectly, can influence creativity by impacting visual perceptions. In Lukersmith and Burgess's (2013) [2] study, healthcare workers reported that calming colors in their work environment—like those on painted walls or furniture—supported their creativity or creative potential within the workspace. Dul and Ceylan (2011) [64], based on a questionnaire survey involving 30 companies, found that the presence of inspiring decorative colors in the physical work environment fosters employees' autonomy in their work, subsequently fostering creativity in their tasks.

The layout of a workspace establishes spatial boundaries, reconfiguring building space and consequently impacting spatial accessibility and visibility. Changes in these boundaries can influence the perception of spatial accessibility and visibility. The visibility of workstations can significantly influence users' sense of psychological safety. Limited visibility tends to have a detrimental effect on communication and collaboration (Stryker et al.,

2012) [67]. Contrary to common perception, open-plan workspaces do not necessarily foster face-to-face employee communication and interaction (Bernstein and Turban, 2018) [68]. Employees situated in neighboring workspaces are more inclined to interact when within each other's shared field of view (Peponis et al., 2007) [66], potentially fostering communicative innovation. In Lukersmith and Burgess's study (2013) [2], spatial privacy emerged as the most influential factor among physical elements like interior decoration and the acoustic environment, significantly affecting creativity according to a questionnaire analysis. Motalebi and Parvaneh (2021) [70], in their focus on the indoor spatial elements impacting artists' creativity, discovered through questionnaires and interviews with 40 artists that privately customized spaces induce relaxation and enhance creative thinking ability. They posited that personalization and a sense of security emerge as two pivotal characteristics of an innovative space conducive to creative work. Kristensen (2004) [69] illustrated via a case study that personal workstations play a central role in influencing the initial stage of innovation, highlighting the importance of catering to individual activity needs.

Across various phases of innovation, individuals exhibit diverse behaviors and, consequently, necessitate distinct acoustic environments. For instance, certain employees necessitate a tranquil environment to concentrate, while others thrive in a communicative atmosphere conducive to innovation promotion. Noise within office settings hampers communication efficiency, undermines organizational cohesion, and disrupts the thought processes crucial to the innovation journey (Clements-Croome, 2006) [71]. Martens (2011) [72] identified through interviews with creative individuals that noise could potentially hinder their creative work. However, the appropriate ambiance of work-related sounds can stimulate creativity. Lukersmith and Burgess (2013) [2], in a questionnaire study involving healthcare workers, identified sound within the work environment as a prospective element for fostering creativity. They suggested various ways to improve the sound environment, such as incorporating sound barriers, employing damping features on flooring surfaces, and providing designated spaces for background or mood music.

### 4.1.3. Comfort

Comfort in the physical environment enhances people's work experience, which in turn promotes innovation. Comfort includes spacious context, comfortable furniture, nature closeness, and light, providing comfort from the perspective of human visual and physical experience (Table 7).

**Table 7.** Comfort of the physical environment affecting innovation.

| Primary Element | Secondary Element | Sources | Nature of Research | Physical Variables | Innovation Variables |
|---|---|---|---|---|---|
| Comfort | Spacious context | Ying et al. (2011) [54] | quantitative | workstation density | collaboration and innovation |
| | | Maryam et al. (2021) [39] | quantitative | workstation density | creative production |
| | | Dian et al. (2022) [38] | quantitative | indoor size | creativity potential |
| | Comfortable furniture | Moultrie et al. (2007) [51] | qualitative | communication tool | creative thinking |
| | Nature closeness | Shibata & Suzuki (2004) [73] | quantitative | indoor plant | creative task |
| | | Atchley et al. (2012) [74] | quantitative | nature | creative inference |
| | | Plambech & Konijnendijk (2015) [75] | qualitative | nature | creativity process |
| | | Ko et al. (2020) [37] | quantitative | view from the window | creative performance |
| | | Chulvi et al. (2020) [76] | quantitative | nature & indoor plant | creative performance |
| | | van den Bogerd et al. (2021) [77] | quantitative | indoor plant | creative cognitive performance |
| | | Yeh et al. (2022) [78] | quantitative | nature | creative performance |
| | Light | Steidle & Werth (2013) [79] | quantitative | light | innovation process |

A generous office space contributes to a comfortable user experience and bolsters job satisfaction (Maryam et al., 2021 [39]; Dian et al., 2022 [38]). Ying et al. (2011) [54]

conducted research incorporating workstation size and workplace space density (the number of employees within a 25-foot radius) as physical variables. They employed a collaboration perception questionnaire and identified a negative correlation between spatial density and collaboration perception. The study highlighted the importance of providing employees with spacious and comfortable workspaces to facilitate collaboration and innovation. In spaces dedicated to innovation, the presence of materials for prototyping and whiteboards for visualizing ideas allows employees to concretize their concepts. This in turn enhances communication comfort and increases the likelihood of innovation (Moultrie et al., 2007) [51].

An ample presence of natural elements can significantly enhance both physical and mental well-being, impacting work innovation through visual perception. Incorporating indoor greenery has been demonstrated to enhance cognitive and creative performance (van den Bogerd et al., 2021 [77]; Shibata & Suzuki, 2004 [73]). Natural landscapes visible through windows can play a crucial role in the preparatory phase of innovation, stimulating visual senses and fostering the generation of novel ideas and heightened creativity (Plambech & Konijnendijk, 2015 [75]; Ko et al., 2020 [37]; Atchley et al., 2012 [74]; Yeh et al., 2022 [78]; Chulvi et al., 2020 [76]). Furthermore, the intensity of light is capable of influencing the conception and incubation of creative ideas by impacting personal perceptions such as attention and mood. This influence is especially prominent during the pre-innovation process (Steidle & Werth, 2013 [79]).

### 4.1.4. Health

The health of the physical environment is reflected in the impact of altered attributes on physical health, which in turn affects work performance and innovation. Healthiness includes two elements, air quality and supply of recreational space, to ensure the physical health of employees (Table 8).

**Table 8.** Health of the physical environment affecting innovation.

| Primary Element | Secondary Element | Sources | Nature of Research | Physical Variables | Innovation Variables |
|---|---|---|---|---|---|
| Health | Air quality | Fang et al. (2004) [80] | quantitative | air quality | creative performance |
| | Supply of recreational space | Hua (2010) [53] | quantitative | number of recreational spaces | support for collaboration |
| | | Sailer (2011) [81] | qualitative | area of recreational space | interaction & creativity |
| | | Candido et al. (2019) [40] | quantitative | area of recreational space | interaction |

Considering the perspective of physical perception, although some current studies have demonstrated the influence of air quality and ventilation on staff's job satisfaction and productivity (Fang et al., 2004) [80], comprehensive research on the nexus between innovation and air quality is yet to be fully developed.

In terms of physical activity, researchers are increasingly examining the impact of open and exercise spaces on employees' physical well-being, communication, and interaction. Hua et al. (2010) [53] evaluated the percentage of leisure and interaction areas in various office buildings along with employees' perceptions of collaboration. Their findings indicated that a smaller percentage of leisure and interaction areas correlated with lower perceived collaboration among employees, thereby hindering effective communication and innovation. Sailer (2011) [81] analyzed leisure and exercise spaces' effects on employees' physical health, communication, and interaction by comparing the spatial layout of a media company's building before and after relocation. This analysis revealed that workplaces with a higher proportion of open spaces exhibited a greater likelihood of episodic communication. Candido et al. (2019) [40] identified that engaging in physical activity during

office hours not only promotes physical fitness but also heightens the likelihood of episodic communication. Such communication patterns, in turn, contribute to fostering innovation.

## 5. Discussion

### 5.1. Physical Environment and Innovation Climate

As early as the early 20th century, the concept of "cognitive maps" was introduced by Tomas (1926) [82], which marked the first instance of associating individual perception with the environment. This connection between subjective perception and the environment laid the foundation for understanding how people perceive their surroundings. As Schneider et al. (2013) [21] articulated, "atmosphere provides a way of accessing tangible things", encompassing both tangible elements and intangible factors within an environment that have the potential to deeply influence human psychology, thereby shaping work behaviors. The impact of the environment on human behavior has increasingly garnered attention from scholars within the field of organizational sociology. However, there remains a dearth of research literature focusing on the perception of the atmosphere and the physical environment through the lens of the architectural discipline. Dul and Ceylan (2011) [64] presented a framework for the effect of personal, social-organizational, and physical factors on employee creativity. Based on this model, we propose a model in which the physical environment and the innovation climate jointly influence the innovation process and, finally, innovation behavior (Figure 4). In contrast to the Dul and Ceylan model, our approach provides a more in-depth exploration of the environment's components and their direct impact on innovation behavior.

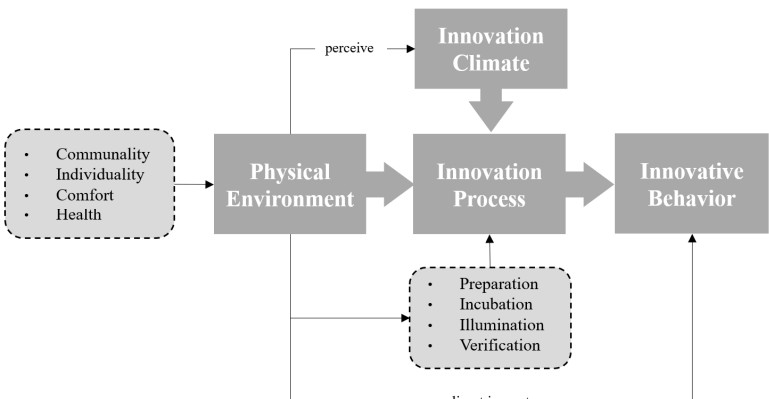

**Figure 4.** A framework for physical environment and innovation climates to promote innovative behavior.

This transition from an intangible culture to a "tangible" climate is pivotal in achieving this goal, as elucidated by Schneider et al. (2013) [21]. A consensus exists among researchers that physical space can serve as a symbol of an organization's values and culture. Organizational culture and climate are the most important organizational factors affecting innovation (Andriopoulos, 2001) [83]. Accordingly, it is stressed that physical environments conducive to fostering innovation should mirror the organization's culture and ethos of innovation (Rafaeli & Vilnai, 2004 [41]; Vilnai et al., 2005 [4]; de Vaujany and Mitev, 2013 [84]; Blomberg & Kallio, 2022 [3]). Notably, less emphasis has been placed on the "tangible" innovation climate within the physical space. Concurrently, the linkage between organizational innovation culture and organizational innovation climate sparks debate within the realm of management science. Ahmed (1998) [42] delineates between innovation culture and innovation climate without explicitly defining their similarities and distinctions. In contrast, Panuwatwanich et al. (2008) [85] identify organizational culture, leadership, and team climate as constituents of the innovation climate. Through empirical analysis, they ascertain that perceptible organizational culture moderates the

other two elements of innovation climate, ultimately fostering innovation and subsequently impacting firm performance.

The interplay between physical space and innovation climate can also be mediated through various other conditions. An illustration of this is the work of Munir and Beh (2019) [86], who establish that organizational innovation climate contributes to knowledge sharing, thereby fostering innovative work behaviors. Additionally, the capacity of communal physical spaces to significantly impact knowledge exchange and sharing in pursuit of collaboration is a consensus among most scholars (Wineman et al., 2009 [33]; Ying et al., 2011 [54]; Salazar & Claudel, 2021 [34]). Another dimension to consider is the influence of the innovation climate's element of interaction frequency, as articulated by Anderson and West (1998) [20], on organizational innovation. This aspect of innovation climate aligns with how spatial layout can impact employee communication, a point underscored by researchers such as Eric and Marry (1986) [30] and Ashkanasy et al. (2014) [87]. These mediating conditions thus reinforce the intricate connections between physical space and innovation climate within organizational contexts.

*5.2. Future Research Directions*

5.2.1. Mechanisms Linking the Innovation Physical Environment and the Innovation Climate

To gain a more comprehensive understanding of the relationship between the physical environment and the climate for innovation, further investigation into the mechanisms through which the physical environment influences the innovation climate is recommended. First, based on the current state of quantitative research on innovation climate, we need to consider how elements of the physical environment for innovation can be systematically quantified. Second, we have analyzed the above and found that there is a certain connection between the physical environment and the innovation climate. This exploration should delve into questions such as: How can innovation climate be quantitatively correlated with the physical environment of innovation in research? Which elements of quantitative research on innovation climate are interrelated with which elements of the physical environment? Specifically, how do elements of the innovation climate and the physical environment work together to influence the various stages of innovation? Addressing these inquiries can deepen our comprehension of the intricate dynamics at play. Considering the present diversity in innovation climate assessment scales and the limited explanations for modifying established scales, this paper advocates for incorporating the influence of physical space into the framework. By integrating spatial factors, future research can enhance the refinement and adjustment of innovation climate scales.

Furthermore, the examination of space's interplay with other managerial components that affect the innovation climate, such as leadership (Lukersmith and Burgess-Limerick, 2013 [2]), could provide a more comprehensive insight. Leadership not only reflects an organization's culture but also its rules and guidelines. Another avenue to explore is the differentiation in the impact of various types of physical environments on the innovation climate. This endeavor can offer practical guidance for cultivating an innovation-friendly climate within organizations. Understanding the nuanced effects of different spatial setups on the innovation climate can aid in crafting tailored strategies for fostering an innovation-oriented atmosphere. In the pursuit of optimizing the innovation climate, organizations are encouraged to capitalize on the advantages offered by the physical environment. By harnessing the potential of the workspace, organizations can amplify their employees' creativity and innovation, thereby propelling the growth and advancement of the entire organization. In sum, the relationship between the physical environment and the innovation climate warrants thorough consideration, necessitating a multidimensional approach to cultivate an optimal innovation climate within organizations.

### 5.2.2. Innovative Symbols of Space and Social Cognition

While organizational culture, normative systems, and values as symbols of physical space have been confirmed by most scholars (Vilnai-Yavetz et al., 2005 [4]; Oksanen & Ståhle, 2013 [52]; de Vaujany & Mitev, 2013 [84]), the current development of firms aiming at sustainable innovation requires scholars to think further about how physical space can symbolize an organization's innovation. When diverse individuals with varying backgrounds and experiences come together within an organizational setting, their shared perception of the physical space contributes to the construction of a collective identity (Brown & Humphreys, 2005 [88]). This shared identity influences how individuals navigate and interact within the organization. In this context, the physical space plays a subtle yet significant role in shaping cultures that foster innovation. Notably, several scholars have confirmed the positive impact of space on the collectivist dimension of innovation culture (Kallio et al., 2015 [89]; Blomberg and Kallio, 2022 [3]). The intricate interplay between the innovation culture represented by the physical space and how this culture is collectively perceived by employees warrants further exploration and investigation. This avenue of inquiry holds the potential to uncover valuable strategies for cultivating an innovation culture that is not just symbolized by the space but also deeply ingrained within the collective identity of the organization.

### 6. Conclusions

Research on the relationship between the physical environment and innovative behavior has historically centered around the field of architecture, with a focus on workspace design (Elsbach and Bechky, 2007) [90]. However, in recent years, this topic has gained recognition and attention from sociological research as well. The existing insights drawn from the literature above offer valuable conclusions about the nexus between innovation and the physical environment. First, the physical environment should be tailored to match the various stages of the innovation process (Haner, 2005 [62]; Pittaway et al., 2019 [91]). For instance, natural environments have notably impacted the preparatory and incubation stages of innovation (Plambech & Konijnendijk, 2015) [75]. Second, architects should design physical environments with different preferred attributes (personalized or communal) depending on the type of work being done. Third, communication and collaboration are essential drivers of innovation within organizations, and the design of physical spaces can significantly influence the occurrence and effectiveness of these interactions (Sailer, 2011) [81].

This paper offers several contributions. First, we propose a ternary framework to guide the analysis of the linkages between innovation, the physical environment, and the climate environment. Second, we inductively propose four elements (communality, personalization, comfort, and healthiness) of the physical environment that affect organizational operations to better review the literature on physical environment research on innovation. We find that current research focuses on the communal and personalized nature of the physical environment for innovation, and in recent years, it has begun to focus on the impact of nature on human-focused innovation, but the impact of the healthfulness of the physical environment on innovation has yet to be studied in depth. Third, based on the four stages of the innovation process, the hypothesis of the mechanism of influence of the physical environment and of the innovation climate on innovation is proposed. Fourth, we integrate research on the physical environment and innovation climate using organizational culture as a hub. Finally, we propose two directions for future research, hoping to provide some reference value for future research on the integrated environment of sustainable innovation. Innovation is a systemic development of organizations and individuals, and we should focus on long-term sustainability and make innovation a core component of a sustainable state of motion.

**Author Contributions:** Conceptualization, L.P. and R.J.; methodology, R.J.; validation, L.P. and R.J.; data curation, R.J.; writing—original draft preparation, R.J.; writing—review and editing, L.P.; visualization, R.J.; supervision, L.P.; project administration, L.P. All authors have read and agreed to the published version of the manuscript.

**Funding:** This research was supported by the National Natural Science Foundation of China, grant number 51978294.

**Institutional Review Board Statement:** Not applicable.

**Informed Consent Statement:** Not applicable.

**Data Availability Statement:** Not applicable.

**Conflicts of Interest:** The authors declare no conflict of interest.

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
