# Peer review of "Exploring the Interplay of the Physical Environment and Organizational Climate in Innovation"

_sustainability, doi:10.3390/su152015013_

Round 1

Reviewer 1 Report

The paper addresses intriguing questions regarding socio-spatial relationships and disciplinary overlaps between sociology and architecture within the context of innovation research. It recognizes a crucial yet underexplored area of study that is certainly worthwhile to explore and has great promise.

Much of the content discussed in the paper falls into qualitative or quantitative categories. However, the paper's approach and position on these different methods and how they can be effectively integrated is currently still unclear. It provides a broad thematic overview but currently lacks an assessment or comparison of evidence and diverse perspectives and approaches within each theme. 

Moreover, the promised three-dimensional framework for analyzing innovative behavior is never fully explained or implemented. There is still a significant need to align the claims made in the introduction with the subsequent content and findings. Currently, these claims appear disjointed and lack the analysis or synthesis of coherent, overarching arguments.

1 Introduction 

The introduction effectively outlines a research problem and objectives. However, these do not fully match what follows. It is essential to clarify from the outset that this paper primarily constitutes a systematic literature review and does not test the proposed three-dimensional analytical framework. This clarification is necessary to prevent confusion and misplaced expectations.

Additionally, more emphasis should be placed on explaining the three-dimensional framework for analyzing innovative behavior. The terms and concepts introduced in Figure 2 require better introduction and justification. The gap between the problem statement and the proposed three-step analysis needs to be bridged. This includes providing clear definitions of key terms, such as "symbolic," and better identifying and, importantly, synthesising the relevant bodies of literature (e.g., social cognitive theory, architectural space theory, and innovation theory) connected to this paper.

2 Innovation and Organizational Environment 

Section 2 attempts to define relevant terms but does so without sufficient justification. It is necessary to provide a rationale for selecting specific definitions over others. For instance, the paper should clarify which innovation environments are considered and why certain types are excluded. Given the focus on architecture and design, this distinction is significant. Despite the initial emphasis on an architectural perspective, the paper appears to engage only minimally with architectural literature.

Furthermore, it is unclear where the terms used in Figure 3 originate, why they are appropriate, and how they relate to each other.

This section largely surveys relevant literature but should be strengthened by clearly indicating which references are pertinent and why. While summarizing others' work, the connections to this paper's position are not as evident as they should be. It is crucial to clarify how this paper builds upon and advances the work of others.

3 Method 

The explanation of the steps involved in selecting and analyzing sources is provided. However, there is insufficient discussion regarding why certain keywords were chosen as meaningful and why others were omitted. The absence of some key terms highlighted in previous sections is surprising. Additionally, the paper does not clarify how the remaining sources were analyzed.

4 Results 

The results should be presented in more structured thematic groups, accompanied by systematic quantification and qualification. Currently, the section combines summaries of arguments by others with discussions of results. The paper should be edited to establish a distinct results section and, if necessary, a discussion section, or these elements should be more explicitly integrated here.

Overall, there is a need for better synthesis, as the section primarily summarizes various works without offering a clear intellectual and analytical framework – or at least explaining the effectiveness of the order of themes currently used.

The categorization, terminology, and discussion are at times unclear and lack rigor. For example, the term "communality" appears to refer mainly to physical proximity, with limited discussion on its connection to communication and its relevance to innovation. This raises broader questions about the overall terminology and conceptual clarity of the paper.

Similarly, the concept of "personalization" seems to overlap with other issues, such as privacy and work habits, without clear definitions and analysis.

The section on "comfort" appears to focus on size rather than environmental comfort, without adequate explanation for this choice.

The use of "healthiness" as a term requires scrutiny, as this section discusses aspects related to nature, physical activity, but lacks clarity on their connection to innovation.

Innovation Climate 

The source and rationale for the proposed model in Figure 5 require better integration within the preceding literature review. An analysis justifying its use and an explanation of its utility and improvement over existing models or approaches are needed.

5 Future Research Directions 

The connection between these future research directions and the literature review in this paper is not fully explained. More work is needed to establish a coherent connection. Some questions and directions might have been more appropriately addressed at the beginning of the paper, rather than as future research questions based on others' work.

The three questions concerning the connection between physical space, innovation climate, and innovation behavior, which are presented as future research questions, should have been explored within this paper, given the introduction and abstract. Their presentation as future research questions is somewhat disappointing. At it currently stands both the abstract and introduction will need major revisions to ensure it aligns with the contents and findings presented.

6 Conclusion 

The origin and evidence supporting the two concluding claims regarding architecture lack an obvious connection to the results presented in the paper. These claims appear to be insights from other authors and need further justification that relates to what is specifically presented in this paper.

The claim regarding the proposal of four elements affecting organizational operations also requires clarification. The paper should elaborate on how it improves the review of physical environment research on innovation and what novel conclusions it draws.

In conclusion, the paper requires substantial revisions to address the issues outlined above and enhance its clarity, coherence, and rigor.

Overall, the quality of writing is good. I feel that the main issue lies in the use of terminology in relation to the conceptualization of the paper, which does not always seem appropriate. 

Author Response

Thank you very much for your encouragement and revision suggestions, your suggestions have greatly improved our paper and we have learned a lot about review writing from them, thank you very much. Here is our response to your revision suggestions:

We have substantially revised the structure and content of the article according to your suggestions, focusing on the abstract and introduction in pursuit of coherence throughout the text. We define each term in detail, based on the literature. This matter is very important and your tips are very much appreciated.

1.Introduction

Thank you very much for your suggestion, we have revised the general statement of the content of the article, emphasizing the literature review part of the paper and explaining in detail the three-dimensional framework for analyzing innovation behaviors in the latter part of the paper

2.Researching the Challenges of Innovation Environments

We revise and explain the content of the three-dimensional framework and match the structure of Chapter 2 to the three elements of the framework. We added an introduction to secondary elements that arise from the intersection of the elements, such as the innovation climate. Innovation, innovation behavior, social environment, organizational climate, physical environment, and the connection between physical environment symbolism and social environment are described in detail. The intersection of innovation and physical environment will be explored in detail in the literature review below.

We have illustrated the sources of the constituent elements of the physical environment, defining the meaning of each element for the literature review in Chapter 4 and repeated again in Chapter 4 to ensure the rigor of the article.

3.Method

We have added and revised the reasons for the selection of keywords and other sources based on your suggestions.

4.Result

Thank you very much, your suggestions have greatly improved the coherence and rigor of this chapter. We have added a consolidated table of elements of the physical environment that influence innovation and explained the current order of the elements.
In each subsection, we added definitions of each element, emphasizing the link between the element and innovation. We integrated the results of the screening of the literature into the table to make the article clearer.

5.Discussion

We have created a discussion section that explores the link between innovative physical environments and innovation climates, based on the symbolism of physical environments in Chapter 2 and the literature summarized in Chapter 4 on innovative physical environments. Based on the exploration, possible directions for future research are suggested.
Based on your suggestions, we have added links between the future directions and the above, and revised the future research directions to further suggest more detailed directions.

6.Conclusion

Based on your suggestions, we have strengthened the link between the concluding claims and the above by adding explanations of the claims of the four elements of the innovative physical environment, improvements in the literature review, and novel conclusions.

Once again, I sincerely thank you for your patient and detailed revision suggestions, which have not only greatly improved this review, but have also allowed us to learn and progress much more, which is much appreciated!

Reviewer 2 Report

The paper is in general well written. It is a very precise and accurate review on a relevant topic. English is in general good. In my opinion section 4.1.2, 4.1.3 etc where can be improved with some images as, as it is, should result a little boring to read.

Please pay attention to some misprint, such as line 107, line 128, line 136, TABLE 3 caption (line 426), line 532

Author Response

Thank you very much for your encouragement and revision suggestions, which have greatly improved our paper. Here are our responses to your revision suggestions:
1. thank you for your suggestion, we have added tables in all 4.1.1-4.1.4 for readers' convenience.
2. thank you very much for your careful checking, we have revised the writing errors you have suggested, you can check at line 166, line 209-210, line 619.

Reviewer 3 Report

The manuscript is well written as a review paper and offers readers a good source of a considerable number of published works on an important topic. While the title refers to sustainable innovation, the review, mostly is about innovation. Does the word sustainability deserve to be in the title, I am wondering.

The text is, on the whole, well written, yet there is a small number of minor issues that need attention. These were highlighted, or commented upon (see embedded comments in the attached pdf of the manuscript. 

Author Response

Thank you very much for your encouragement and revision suggestions, which have greatly improved our paper. Here are our responses to your revision suggestions:
1. we removed the word sustainable to keep the theme of the essay consistent
2. we added the citation for this sentence, which you can check at line38-40
3. as per your suggestion, we checked the whole article to avoid repetition "organizations"
4. as per your suggestion, we have modified the altered figure, which you can check at figure 2 on page 21
5. we changed the words and grammatical errors you suggested and checked the full text
6. we changed healthiness to health, a dimension intended to explore the impact of the physical environment on people's physical health.
7. this was a mistake on our part and we have changed the name of the table, which you can check in Table 1 on page 21.
8. public in Figure 5 has been changed to communality and other words have been harmonized, thank you for checking.
